



# Spring and summer time ozone and solar ultraviolet radiation variations over Cape Point, South Africa

D. Jean du Preez[1,2], Jelena V. Ajtić[3], Hassan Bencherif[4], Nelson Bègue[4] and Caradee Y. Wright[1,5]

[1*]Department Geography, Geoinformatics and Meteorology, University of Pretoria, Pretoria, 0002, South Africa
5  [2]Research Department, South African Weather Service, Pretoria, 0181, South Africa
[3]Faculty of Veterinary Medicine, University of Belgrade, Bulevar oslobodjenja 18, 11000 Belgrade, Serbia
[4]Université de La Réunion, Laboratoire de l'Atmosphère et des Cyclones, UMR 8105, 15 Avenue René Cassin, CS 92003, Saint-Denis, Cedex, Réunion, France
[5]Environment and Health Research Unit, South African Medical Research Council, Pretoria, 0001, South Africa

*Correspondence to*: D. Jean du Preez (dupreez.jd@gmail.com)

**Abstract.** The correlation between solar ultraviolet radiation (UV) and atmospheric ozone is well understood. Decreased stratospheric ozone levels which led to increased solar UV radiation levels at the surface have been recorded. These increased levels of solar UV radiation have potential negative impacts on public health. This study was done to determine whether or not the break-up of the Antarctic ozone hole has an impact on stratospheric columnar ozone (SCO) concentrations and resulting ambient solar UV-B radiation levels at Cape Point, South Africa. At Cape Point, the strongest anti-correlation on clear-sky days was found at solar zenith angle 20° with exponential fit $R^2$ values of 0.71 and 0.66 for total ozone column and SCO, respectively. An average radiation amplification factor of 0.92 was found and the largest decrease in ozone levels occurred during September months. The MIMIOSA-CHIM model showed that the polar vortex had a limited effect on ozone levels at 435 - 440 K for September and 600 K over Cape Point during November. Tropical air-masses more frequently affect the study site, and this requires further investigation.

**Keywords: UV radiation; clear-sky; stratospheric-ozone**

## 1. Introduction

Solar ultraviolet (UV) radiation is a part of the electromagnetic spectrum of energy emitted by the Sun (Diffey, 2002). Solar
25  UV radiation comprises a wavelength band of 100 - 400 nm, however, not all wavelengths reach the Earth. Solar UV radiation is divided into UV-A, UV-B and UV-C bands depending on the wavelength. The UV-C and UV-A bands cover the shortest and longest wavelengths, respectively. The UV-B part of the spectrum spans a wavelength range between 280 - 315 nm (WHO, 2017). The reason behind this sub-division of UV radiation is a large variation in biological effects related to the different wavelengths (Diffey, 2002). Moreover, an interaction of different UV bands with the atmospheric constituents, results in an
30  altered UV radiation reaching the surface: all UV-C and ~90 % of UV-B radiation is absorbed, while the UV-A band is mostly unaffected (WHO, 2017). The amount of solar UV-B radiation at the surface of the Earth is largely impacted by the amount



of atmospheric ozone (Lucas & Ponsonby, 2002), but also several other factors, such as altitude, solar zenith angle (SZA), latitude, and pollution (WHO, 2017). The SZA has a significant impact on the amount of surface solar UV-B radiation (McKenzie, et al., 1996). Under clear-sky conditions and low pollutions levels, atmospheric ozone (of which approximately 90% is found in the stratosphere) absorbs solar UV-B radiation (Fahey & Hegglin, 2011). A study in the south of Brazil found

a strong anti-correlation between ozone and solar UV-B radiation on clear-sky days using fixed SZAs (Guarnieri, et al., 2004).

Anthropogenic and natural factors can cause changes in the amount of atmospheric ozone. Unlike natural ozone variability which is mostly of a seasonal nature and therefore has a reversible character, human activities, such as release of ozone-depleting substances, have led to a long-term ozone decline in a greater part of the atmosphere (Bais, et al., 2015), and, in turn,

to higher levels of solar UV-B radiation at the Earth's surface (Fahey & Hegglin, 2011). An outstanding example of ozone depletion is the formation of the Antarctic ozone hole, a phenomenon discovered in the 1980s (Farman, et al., 1985). Each austral spring, a severe ozone depletion occurs under the unique conditions in the Antarctic polar vortex, decreasing total ozone column (TOC) below 220 Dobson Units (DU), a threshold defining the ozone hole.

The Antarctic ozone hole has been extensively studied (WMO, 2011). Apart from its direct influence on the ozone amounts in the Southern Hemisphere (Ajtić, et al., 2004; de Laat, et al., 2010) the Antarctic ozone hole affects a wide range of atmospheric phenomena as well as climate of the Southern Hemisphere. For example, ozone depletion over Antarctica has altered atmospheric circulation, temperature and precipitation patterns in the Southern Hemisphere during the austral spring and summer (Brönnimann, et al., 2017; Bandoro, et al., 2014). Another notable consequence of decreased atmospheric ozone is an

increase in solar UV radiation at the surface of the Earth, which has been supported by experimental evidence (Herman & McKenzie, 1998). This anti-correlation and association with the Antarctic ozone hole has been confirmed at Lauder, New Zealand (McKenzie, et al., 1999).

Our analysis investigated the anti-correlation between ozone and solar UV-B radiation over the Western Cape Province, South

Africa. The objectives in our study were: 1) to determine the climatology of solar UV-B radiation and  the climatology of TOC and stratospheric column ozone (SCO) for Cape Point, South Africa; 2)  to determine clear-sky days for Cape Point and use them to analyse the anti-correlation between solar UV-B radiation and TOC, on one hand, and solar UV-B radiation and SCO, on the other hand; 3) to identify low TOC and SCO events at Cape Point focusing on spring and summer months; 4) to use a transport model to determine the origin of ozone-poor air observed during low-ozone events;  and 5) to explore whether the

Antarctic ozone hole influenced these low-ozone events at Cape Point. To the best of our knowledge these objectives, in a South African context, and in relation to increased solar UV-B radiation over South Africa directly related to the Antarctic ozone depletion, have not been studied before.



## 2. Data and methods

### 2.1 UV data

The study site was Cape Point (34.35 °S, 18.50 °E, 230 m a.s.l.) a weather station in the Western Cape, (Fig. 1) South Africa. The station is one of the World Meteorological Organization (WMO) Global Atmospheric Watch (GAW) baseline monitoring

sites (GAW, 2015). It is located around 60 km south of Cape Town and is considered free of air pollution (Slemr, et al., 2008). Since aerosols can have a pronounced effect on the amount of UV radiation reaching the surface (Bais, et al., 2015), our choice of Cape Point offers a setting in which a modification of the UV-B radiation by aerosols can be overlooked.

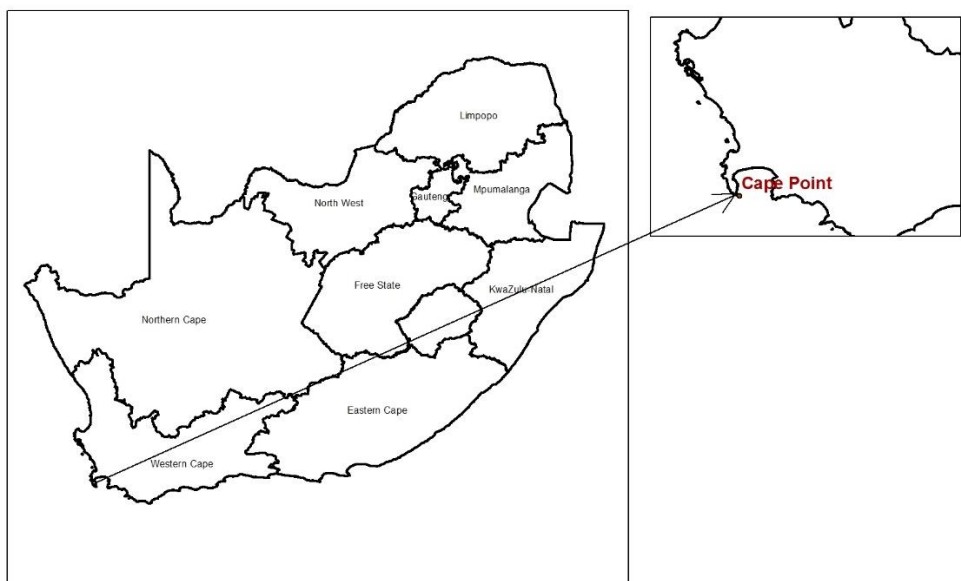

**Figure 1:** The map of South Africa and the location of the SAWS Cape Point Weather station in the Western Cape (L. Thobela, SAWS
2018).

Hourly solar UV-B radiation data were obtained from the South Africa Weather Service (SAWS) for Cape Point station for the period 2007 - 2016. The solar UV-B radiation measurements were made with the Solar Light Model Biometer 501 Radiometer. The biometer measures solar UV radiation with a wavelength of 280 - 320 nm. The measured solar UV radiation

is proportional to the analogue voltage output from the biometer with a controlled internal temperature (Solarlight, 2014). Measurements are given in Minimal Erythemal Dose (MED) units where 1 MED is defined by SAWS as 210 Jm$^{-2}$ and any incorrect or missing values were indicated in the dataset. During October 2016, the measured MED values exceeded the expected values and were corrected with a correction factor as recommend by the SAWS. Despite periods of missing data during the study years, there were 3 129 days of useable solar UV-B data for Cape Point.

Solar UV-B radiation values in MED were converted to UV index (*UVI*) values using Equation (1).



$$UVI = \left(\frac{MED \times 210}{3600}\right) \times 40 \hspace{6cm} (1)$$

where UVI is the Ultraviolet (UV) Index (WHO, 2017) and MED is Minimal Erythemal Dose units.

## 2.2 Column ozone data

TOC and SCO data were obtained for 2007 – 2016 (inclusive) for the grid area which was bound by the following coordinates
- West: 16.5 °E, South: 36.35 °S, East: 20.6 °E, North: 31.98 °S. This grid area limited the TOC and SCO data to the area directly above Cape Point. The daily TOC data were measured with the Ozone Monitoring Instrument (OMI) on NASA's Aura satellite. OMI has a spatial resolution of 0.25° which results in a ground resolution at nadir with a range of 13 km x 24 km to 13 km x 48 km (Levelt, et al., 2006).

The daily SCO data were measured with the Microwave Limb Sounder (MLS) instrument on NASA's Aura satellite. The MLS ozone data consisted of ozone profiles at 55 pressure levels and SCO values up to the thermal tropopause. The thermal tropopause is determined by the temperature data taken by the MLS instrument. The ozone profiles were used between 261 - 0.02 hPa (Livesey, et al., 2017). The daily SCO values were extracted from the MLS data files.

## 2.3 Transport model

The Mesoscale Isentropic Transport Model of Stratospheric Ozone by Advection and Chemistry (Modèle Isentropique du transport Méso-échelle de l'ozone stratosphérique par advection avec CHIMIE or MIMOSA-CHIM) was used to identify the source of ozone-poor air above Cape Point. MIMOSA-CHIM results from the off-line coupling of the MIMOSA dynamical model (Hauchecorne, et al., 2002) from the Reactive Processes Ruling the Ozone Budget in the Stratosphere (REPROBUS) chemistry model (Lefèvre, et al., 1994). The ability of MIMOSA-CHIM to simulate and analyze the transport of stratospheric
air-masses has been highlighted in several previous studies over the polar regions (Kuttippurath, et al., 2013; Kuttippurath, et al., 2015; Tripathi, et al., 2007; Semane, et al., 2006; Marchand, et al., 2003). The dynamical component of the model is forced by meteorological data such as wind, temperature, and pressure fields from the European Centre for Medium-Range Weather Forecasts (EMCWF) daily analyses. The dynamical component of potential vorticity (PV) was used to trace the origin of ozone-poor air-masses. PV can be used as a quasi-passive tracer when diabatic and frictional terms are small. Therefore, over
short periods of time, PV is conserved on isentropic surfaces following the motion (Holton & Hakim, 2013). A spatial area from 10 °N to 90 °S was used for the model with a 1° x 1° resolution. The model has stratospheric isentropic levels ranging from 350 - 950 K. The MIMOSA-CHIM model created an output file for every 6 hours and simulations were initialised to run for approximately 14 days prior to each low-ozone event to account for the model spin-up period. The PV maps were analysed at isentropic levels that correspond to 18 km, 20 km and 24 km above ground level, thus covering the lower part of the ozone
layer (Sivakumar & Ogunniyi, 2017).



## 2.4 Method

### 2.4.1 Climatologies

The hourly average for each day in a specific month across the 10-year period was used to determine the solar UV radiation climatology at Cape Point. All days were used to determine the climatology, regardless of cloud conditions. The TOC and

SCO climatologies were calculated using monthly averages. The climatologies and patterns analysed used all available days with data.

### 2.4.2 Determination of clear-sky days

Clouds are an important factor that attenuate or enhance the solar UV-B radiation at the Earth's surface (Bais, et al., 2015). Thus, determining cloud-free days is a step towards removing a contribution of all factors except amount of atmospheric ozone.

As shown by (McKenzie, et al., 1991) stronger correlation between ozone and solar UV-B radiation may be obtained if the days with clouds are removed.

To remove the effect of cloud cover on solar UV-B radiation, we used a clear-sky determination method by Bodeker and McKenzie (1996). Days with solar UV-B measurements, TOC and SCO data were divided into seasons: summer – December,

January, February (DJF), autumn – March, April, May (MAM), winter – June, July, August (JJA) and spring – September, October, November (SON). Clear-sky days were determined using three different tests.

The first test only considered the daily linear correlation between the solar UVI values measured before solar noon and the values after solar noon. Solar noon was determined as the hour interval with the lowest SZA value. Days with a linear

correlation below 0.8 in the DJF, MAM and SON seasons were removed and were considered to be cloudy days. The first test was not performed on the JJA season, when the UVI values were low, and the correlation values were also very low.

The second test looked for a monotonic increase before solar noon and a monotonic decrease after solar noon for each day. On clear-sky days, UVI values before and after solar noon should monotonically increase and decrease, respectively. If

monotonicity did not hold for the UVI values on a specific day, it was assumed that there was some cloud present on that day. The monotonicity test was performed for all seasons.

The third test removed days when the UVI values did not reach a threshold maximum value. This test was applied to all seasons. The threshold was determined as a value of 1.5 standard deviations (1.5 STD) below the UVI monthly average. The

monthly average and standard deviations were determined from the solar UV-B radiation climatology for Cape Point.



The determination of clear-sky days was tested on a solar UV-B radiation dataset from the Cape Town weather station as this station has cloud cover data which could be used to validate the results of the clear-sky tests. The validation was done using the daily 06h00 UTC and 12h00 UTC cloud cover observations. Two randomly selected years were used for the validation and it was found that when the cloud cover observations indicated that more than 4/8ths of cloud were present, the clear-sky

tests had determined these days as cloudy. Furthermore, the diurnal radiometric curves from another year were examined and found that determined clear-sky days' radiometric curves closely followed the expected diurnal radiometric curve. This validation confirmed that the clear-sky tests did remove approximately 87% of cloudy days. Approximately 500 days were determined to be clear-sky days that had UVI, TOC and SCO data and were used for the analyses.

### 2.4.3 Correlations

In addition to removing aerosols by choosing an air pollution free site and alleviating cloud effects by looking only at the clear-sky days, the correlation between solar UV-B radiation and ozone can be better observed when controlling SZA (Booth & Madronich, 1994). To calculate ten-minute SZAs for Cape Point, we applied an online tool, Measurement and Instrument Data Centre, Solar Position Calculator (MIDC SPA)(https://midcdmz.nrel.gov/solpos/spa.html accessed February 2017), which uses the date, time and location of the site of interest and has an accuracy of +/-0.0003° (Reda & Andreas, 2008).

The strength of the correlation between ozone (TOC and SCO data) and solar UV-B radiation was determined using the first order exponential fit (Guarnieri, et al., 2004):

$$y = ae^{bx} \tag{2}$$

where $y = UVI$ and $x = $ ozone values (TOC or SCO).

The significance of the goodness of fit was determined for a 95% confidence interval. The log UVI (y-axis) values were taken in order to test if the goodness of fit $R^2$ values of the exponential fits were statistically significant (Hazarika, 2013).

### 2.4.4 Radiation Amplification Factor

The Radiation Amplification Factor (RAF) describes a relationship between ozone values and solar UV-B radiation (Booth &
Madronich, 1994). The RAF was introduced as a quantification of the effect that decreased ozone concentrations have on solar UV-B radiation levels. The RAF is a unitless coefficient of sensitivity and here we used its definition given by Booth & Madronich (1994) in Equation 3. The RAF value at fixed SZAs was calculated using a specific clear-sky day compared to another random clear-sky day from a different year (Booth & Madronich, 1994).

$$RAF = \ln(\frac{O3}{O3'})/\ln(\frac{UVI'}{UVI}) \tag{3}$$



where $O_3$ and $O_3$' are the first and second ozone values and *UVI* and *UVI*' are the first and second UVI measurements, respectively.

### 2.4.5 Low-ozone days

Days of low TOC and SCO values were determined from the set of clear-sky days, but only during spring and summer seasons, when solar UV radiation levels are highest. Days of low TOC values might not have had low SCO values and vice versa. Low TOC and SCO days were determined as days when the respective values were below 1.5 STD from the mean as determined in the climatology analyses (Schuch, et al., 2015).

We then used the MIMOSA-CHIM model to identify whether the origin of ozone-poor air-masses was from the polar region. The maps of advected PV from MIMOSA-CHIM were used to identify the source of ozone-poor air parcels over the study area for low-ozone days that had been identified.

## 3. Results and discussion

### 3.1 Climatologies and trends

### 3.1.1 Solar UV-B climatology

The monthly means of the solar UV-B radiation for Cape Point during 2007–2016 were calculated as a function of time of day and month of the year (Fig. 2). This climatology provides a reliable baseline against which observations can be compared and reveals the general patterns of the UV-B signal recorded at the surface over the investigated 10-year period.

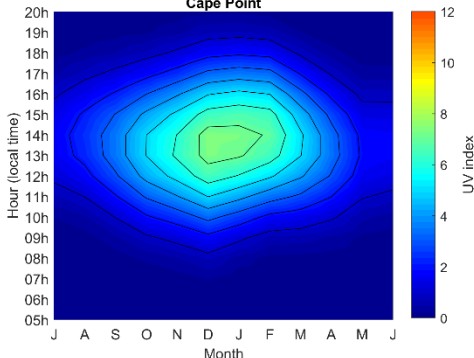

**Figure 2: The *UVI* climatology for all sky conditions at Cape Point. The *x*-axis starts with the month of July and ends with June.**

At Cape Point the maximum UVI occurred between December and January of each year, and typically between 13h00 and 15h00 UTC+2 daily, with the maximum value of approximately 8 (Fig.2). The periods of maximum (DJF) and minimum (JJA) solar UV-B radiation at Cape Point are as expected for a site in the Southern Hemisphere and are similar to those found in a



study at four other South African sites, namely Pretoria, Durban, De Aar, and Port Elisabeth and Cape Town (Wright, et al., 2011; Cadet, et al., 2017).

### 3.1.2 Ozone climatologies

TOC (303.4 DU) and SCO (273.1 DU) values peaked during September and decreased to a minimum in February for SCO
(232.65 DU) and April for TOC (254.49 DU) (Fig. 3). The variations in TOC and SCO are largest at the maximum values and smallest at the minimum values. Over Irene in Pretoria the greatest variation in SCO was seen during spring (Paul, et al., 1998), which is in agreement with our results. It is suggested that this variability in TOC is due to the movement of midlatitude weather systems which move further north during the Southern Hemisphere winter (Diab, et al., 1992). The climatology of TOC over South Africa is mainly affected by atmospheric dynamics rather than by the effects of atmospheric chemistry
(Bodeker & Scourfield, 1998).

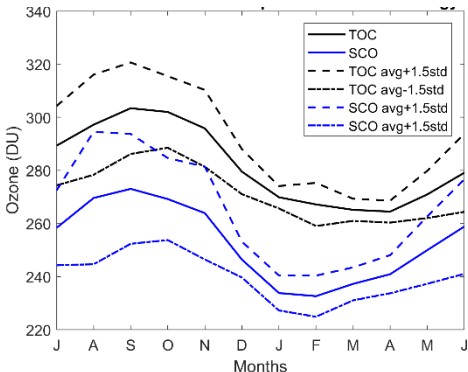

**Figure 3: Monthly means and +/- 1.5 STD for total ozone column and stratospheric column ozone starting in July and ending in June.**

The increase in TOC values during the winter months and the maximum during spring months are due to an ozone-rich midlatitude ridge that forms on the equator-side of the polar vortex. The ridge is a result of a distorted meridional flow caused by the Antarctic polar vortex that forms in late autumn. The vortex prevents poleward transport of the air, and thus allows for a build-up of ozone-rich air in midlatitude. The lower TOC values over summer could be due to the dilution effect of ozone-poor air from the Antarctic ozone hole. The dilution effect occurs when the vortex breaks-up (Bodeker & Scourfield, 1998;
Ajtić, et al., 2004).

### 3.2 Correlation between ozone and solar UV-B radiation

The first order exponential goodness of fit $R^2$ values and the RAF values at fixed SZAs (Table 1) describe the anti-correlation between atmospheric ozone and UVI.

**Table 1. The correlation statistics for ozone and UVI at Cape Point on clear-sky days.**



| SZA (°) | TOC: $R^2$ Expo fit | SCO: $R^2$ Expo fit | RAF |
|---|---|---|---|
| 15 | 0.60* | 0.54* | 0.50 |
| 20 | 0.71* | 0.66* | 0.57 |
| 25 | 0.41* | 0.30* | 0.24 |
| 30 | 0.50* | 0.44* | 0.34 |
| 35 | 0.22* | 0.10* | 0.18 |
| 40 | 0.01 | 0.00 | 1.49 |
| 45 | 0.01 | 0.00 | 3.09 |
| Average | | | 0.92 |

Note: *-indicates which goodness of fit $R^2$ values were statistically significant at a 95% confidence interval.

The strongest statistically significant anti-correlation was found at a fixed SZA 20° for both TOC and SCO. The goodness of fit $R^2$ values at fixed SZAs 40° and 45° were found not to be statically significant. In general, the strength of the goodness of fit $R^2$ tended to decrease as the size of the SZA increased.

For statistically significant goodness of fit $R^2$ values between TOC and UVI, the $R^2$ values were 22.4 – 71.0% (SZA 15° - 35°) and explained the variation of solar UV-B radiation with variations in TOC at fixed SZAs on clear-sky days for Cape Point. For statistically significant goodness of fit $R^2$ values between SCO and UVI the percentage $R^2$ values were 10.4 - 66.4% (SZA 15° - 35°) where this explains the variation of solar UV-B radiation with variations in SCO at fixed SZAs on clear-sky days for Cape Point. The lowest RAF value was seen at a fixed SZA 25° and the highest at 45°. The average RAF value across all

SZAs was 0.9.

In this study the first order exponential fit was used to describe the anti-correlation between ozone and solar UV radiation as in some instance this is best described with a non-linear fit (Guarnieri, et al., 2004). A study on the anti-correlation between

solar UV-B irradiance and TOC in southern Brazil found that the percentage of the $R^2$ values for exponential fits (66.0 -85.0%) explained the variations in solar UV-B irradiance due to TOC variations on clear-sky days at the same fixed SZA categories used in our study (Guarnieri, et al., 2004). The exponential $R^2$ values of TOC found at Cape Point at a fixed SZA of 20° were close to $R^2$ vales found for southern Brazil. The exponential fits found for TOC and SCO at Cape Point correspond best at smaller SZAs similar to other studies (Guarnieri, et al., 2004; Wolfram, et al., 2012). In general, the strength of the anti-

correlation decreases with an increase in SZA and this is found to be true at Cape Point but the decrease starts at smaller SZAs at Cape Point than in other studies such as the one in southern Brazil (Guarnieri, et al., 2004).



RAF values specific to ozone and solar UV studies found in the literature range between 0.79 and 1.7 (Massen, 2013) and increase as the SZA increases (Herman, 2010). The RAF values at Cape Point range from 0.50 to 3.09 with an average of 0.92 across all fixed SZA categories. The RAF values at Cape Point suggest an increasing trend as SZAs increase (Table 1). The differences to the RAF values found here and those found in the literature can be attributed to changes in time and location

(Massen, 2013).

The method used for this study was chosen due to the use of the broadband biometer used to measure solar UV-B radiation between 280 - 320nm as well as the temporal resolution, i.e. 60 minutes, of measurements (Solarlight, 2014). Only clear-sky days were used as clouds can significantly impact solar UV-B radiation measured at the surface (Prause & Scourfield, 2002).

The correlation coefficients found here may be influenced by factors that could not be removed or determined when conducting the data analysis. Salt build-up on the biometer at Cape Point may also have contributed to the accuracy of measurements taken by the biometer. Solar UV-B radiation data with a higher temporal resolution (e.g. 10 minutes) may have provided more data points for the analysis at fixed SZAs. Higher temporal resolution solar UV-B radiation data would have improved the determination of clear-sky days. An improvement on the correlation and RAF values could be made by investigating the

aerosol concentrations over the station.

### 3.3 Low-ozone events

Low-ozone events which occurred during the SON and DJF months were identified from the time series of TOC and SCO data on clear-sky days (Fig. 4). The highest frequency of low TOC and low SCO events occurred during January months and January and December months, respectively.

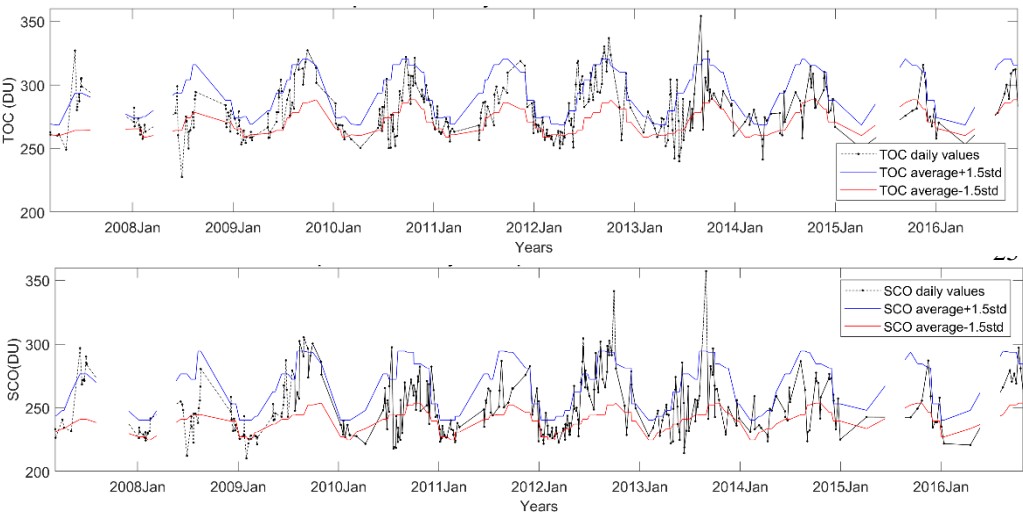

**Figure 4**: *TOC (top) and SCO (bottom) values on clear-sky days over Cape Point and an indication of the average ±1.5 STD limits. Each dot corresponds to a TOC and SCO measurement on a clear-sky day from 2007-2016. Selected areas have been blocked out where large data gaps were presented.*





The low TOC and low SCO events along with the respective percentage decrease in TOC and SCO (Table 2) represent some of the largest decreases that occurred in DJF and SON seasons between 2007 and 2016 on clear-sky days. The DJF seasons of 2009/2010 and 2015/2016 are classified as El Niño years (Climate Prediction Center Internet Team, 2015). During these seasons higher TOC levels are expected over the midlatitude regions (Kalicharran, et al., 1993). From the identified low TOC events at Cape Point, none occurred during El Niño years.

**Table 2. Identified low-ozone events on clear-sky days at Cape Point during spring and summer months and the percentage decrease calculated from the relative climatological monthly mean.**

| Date | TOC (DU) | SCO (DU) | Decrease TOC (%) | Decrease SCO (%) | Increase in UVI (%) |
|---|---|---|---|---|---|
| 30 Jan 2009 | 253.5* | 210.4* | 6.1 | 10.1 | 28.9 |
| 6 Feb 2009 | 253.9* | 222.2* | 5.0 | 4.5 | 26.4 |
| 15 Feb 2009 | 254.6* | 228.3 | 4.7 | 1.9 | 24.7 |
| 28 Feb 2011 | 255.7* | 223.2* | 4.3 | 4.1 | 7.5 |
| 16 Jan 2012 | 268.2 | 221.9* | 0.6 | 5.1 | 9.3 |
| 8 Feb 2012 | 257.0* | 227.5 | 3.8 | 2.2 | 31.2 |
| 13 Nov 2012 | 256.6* | 228.8 | 13.3 | 13.3 | 31.3 |
| 14 Nov 2012 | 261.3* | 234.6 | 11.7 | 11.1 | 38.6 |
| 6 Sep 2013 | 265.0* | 241.3* | 12.7 | 11.6 | 23.3 |
| 9 Nov 2013 | 282.3 | 229.0* | 4.6 | 13.3 | 20.5 |
| 1 Sep 2014 | 274.7* | 223.9* | 9.5 | 18.0 | -1.1 |
| 2 Sep 2014 | 258.4* | 231.2* | 14.9 | 15.3 | -1.5 |
| 9 Sep 2014 | 284.0* | 232.3* | 6.4 | 14.9 | -4.2 |
| 11 Jan 2016 | 270.6 | 221.9* | -0.3 | 5.1 | -4.9 |

Note: *-indicates whether the low-ozone event was due to low TOC and/or low SCO values

This analysis aimed to discuss effects of stratospheric ozone and tropospheric ozone on surface UV-B radiation variations. When TOC and SCO percentage reductions are similar, it means that the effect of stratospheric ozone is dominant. Conversely, when the percentage reduction of TOC is high, and the percentage reduction of SCO is low, the effect of tropospheric ozone is dominant. All of the low-ozone events which occurred during January were due to decreased SCO. A decrease of 10.1% in SCO was recorded on 30 January 2009 with a TOC decrease of 6.1%. During February months we obtained the weakest low



TOC and SCO percentage reduction. Low ozone events that occurred during September were mainly due to stratospheric ozone decreases, with the largest ozone reduction recorded on 01 September 2014 (18% SCO reduction) (Table 2). A similar situation occurred in November when low ozone events are mainly due to decreases in tropospheric and stratospheric ozone.

UVI levels which occurred during low-ozone events within the SON and SJF seasons were compared to the UVI climatology to determine if there was any effect on UVI levels during low-zone events.  At Cape Point, the largest increases in UVI levels were recorded for low-ozone events during November.

In the Southern Hemisphere, during the spring season (SON) low-ozone events are predominately due to the distortion and
filamentation of the Antarctic ozone hole and to the dilution of the associated polar vortex. The dilution effect occurs later in the early summer season, when ozone-poor air-masses from polar regions mix with air-masses from the midlatitudes and result in decreased ozone concentrations (Ajtić, et al., 2004). There are no studies that refer to low-ozone events at Cape Point. Due to the relative proximity of the study area to the Antarctic ozone hole it is expected that some similarities could be found with other Southern Hemisphere sites. In South Africa, a decrease in TOC was observed over Irene (25.9°S, 28.2°E) during May
2002 when TOC levels were 8 – 12 % below normal and at a minimum of 219.0 DU. (Semane, et al., 2006). Tropospheric and stratospheric interaction has been identified using PV as a tracer for stratospheric air-masses. The relative position of the surface high- or low-pressure can result in increases or decreases in TOC. The effect on TOC by weather systems is seasonal dependent (Barsby & Diab, 1995).

Over southern Australia, solar UV-B radiation increased by 40 % during a low SCO event even though it was winter (Gies, et al., 2013). The increased levels of solar UV-B radiation found in this study due to low SCO events are lower than those found at other Southern Hemisphere sites (Gies, et al., 2013; McKenzie, et al., 1999; Abarca, et al., 2002). It is possible that low-ozone events that occurred over the region during 2007-2016 have not been included. These events might have fallen outside the methods used in this study or were not considered due to the availability of solar UV-B radiation, TOC or SCO data.
Moreover, it should be noted that the Cape Point site being located at 34° South, at the southern limit of the tropical stratospheric reservoir. Cape Point can be affected by dynamical and transport processes, and therefore air-masses of different latitude origins can pass over it. Indeed, over our study period from September to February, the obtained low ozone event could be of polar origin, i.e., in relation with the extension and distortion of the polar vortex, or of tropical origin, i.e., in relation with isentropic air-masses transport across the subtropical barrier, as reported by (Semane, et al., 2006; Bencherif, et
al., 2011; Bencherif, et al., 2007). The following sub-section discusses low ozone events with regard to the dynamical situations and origins of air-masses above the study site.



### 3.4 Origin of ozone-poor air

In this section the model results from MIMOSA-CHIM are shown for a selection on low-ozone events. Low-ozone events during January and February were not directly influenced by the Antarctic ozone hole as it was no longer established over Antarctica. The latitude origin of air-masses was classified according to the colour scale on the PV maps. Blue colours indicate

air-masses with relatively high PV values from polar origins, while red colours indicate air-masses with relatively low PV values of tropical origin.

The origin of the air-masses for low-ozone events in January (Table 3) show a general pattern. The PV maps from MIMOSA-CHIM of the low-ozone event on 16 January 2012 (Fig. 5) are shown. This low-ozone event best demonstrates all January

events. During this event SCO were identified to be low (Table 2).

**Table 3. Origin of ozone-poor air at isentropic levels for low ozone events in January.**

| Date | Origin at 425 K | Origin at 475 K | Origin at 600 K |
|------|-----------------|-----------------|-----------------|
| 30 Jan 2009 | Tropical | Midlatitude | Polar |
| 16 Jan 2012 | Tropical | Midlatitude | Polar |
| 11 Jan 2016 | Tropical | Midlatitude | Polar |

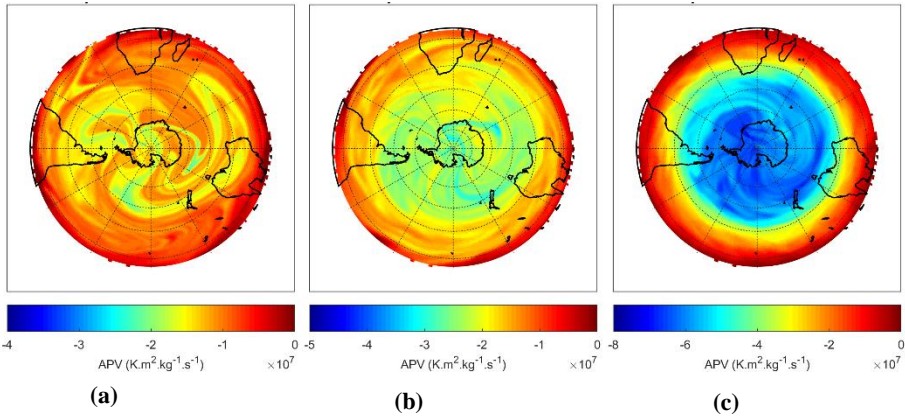

(a)                                (b)                                (c)

**Figure 5:  PV maps from MIMOSA-CHIM at 425 K (a), 475 K (b) and 600 K (c) on 16 January 2012.**

The origin of air-masses for low-ozone events in February (Table 4) show a similar pattern to January events. The PV maps from MIMOSA-CHIM for the low-ozone event on 6 February 2009 (Fig. 6) best demonstrate the situation for February months.



**Table 4. Origin of ozone-poor air at isentropic levels for low ozone events in February.**

| Date | Origin at 425 K | Origin at 475 K | Origin at 600 K |
|---|---|---|---|
| 6 Feb 2009 | Tropical | Midlatitude | Polar |
| 15 Feb 2009 | Tropical | Midlatitude | Polar |
| 28 Feb 2011 | Tropical | Midlatitude | Polar |
| 8 Feb 2012 | Tropical | Midlatitude | Polar |

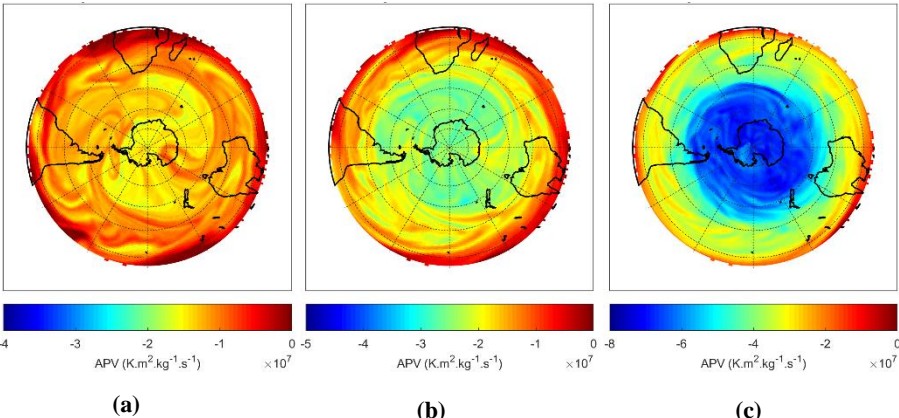

**(a)**       **(b)**       **(c)**

**Figure 6: PV maps from MIMOSA-CHIM at 425 K (a), 475 K (b) and 600 K (c) on 6 February 2009.**

5   The origin of air-masses for low ozone events during September (Table 5) and the PV maps from MIMOSA-CHIM for the low-ozone event on 2 September 2014 (Fig. 7) shows the transport of tropical air-masses southward over the study site. During September months there was less mixing of air-masses across latitudinal boundaries.

**Table 5. Origin of ozone-poor air at isentropic levels for low ozone events in September.**

| Date | Origin at 435 K | Origin at 485 K | Origin at 600 K |
|---|---|---|---|
| 6 Sep 2013 | Tropical | Midlatitude | Midlatitude - Polar |
| 1 Sep 2014 | Tropical | Tropical | Midlatitude - Polar |
| 2 Sep 2014 | Tropical | Tropical | Midlatitude - Polar |
| 9 Sep 2014 | Tropical | Tropical | Midlatitude - Polar |



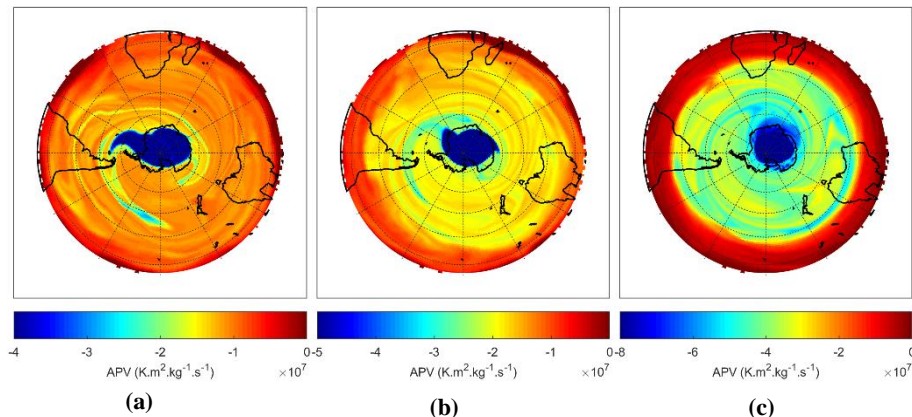

**Figure 7: PV maps from MIMOSA-CHIM at 435 K (a), 485 K (b) and 600 K (c) on 2 September 2014.**

The origin of air-masses for low-zone events in November (Table 6) show that at 600 K polar air-masses do affect the study

5  site but ozone hole is no longer present over Antarctica. The PV maps from MIMOSA-CHIM for the low-ozone event on 14 November 2012 (Fig. 8) best demonstrate the situation for November months.

**Table 6. Origin of ozone-poor air at isentropic levels for low ozone events in November.**

| Date | Origin at 435 K | Origin at 480 K | Origin at 600 K |
|---|---|---|---|
| 13 Nov 2012 | Tropical | Midlatitude | Polar |
| 14 Nov 2012 | Tropical | Midlatitude | Polar |
| 09 Nov 2013 | Tropical | Midlatitude | Polar |

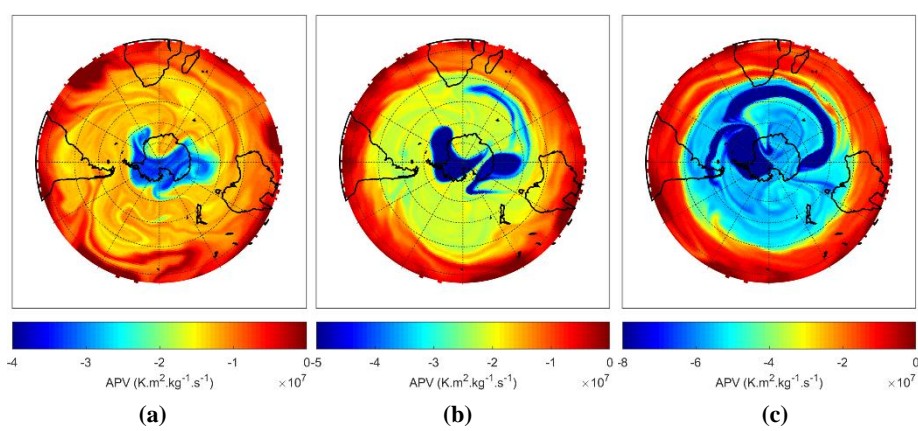



**Figure 8: PV maps from MIMOSA-CHIM at 435 K (a), 485 K (b) and 600 K (c) on 14 November 2012.**

The PV maps from MIMOSA-CHIM suggest that the Antarctic polar vortex air-masses with low-ozone levels have a limited effect on the ozone levels over Cape Point, South Africa. Instead, the study site is largely influenced by ozone-poor air-masses from sub-tropical regions. The effect of these sub-tropical air-masses on ozone concentrations is dependent on isentropic level

and time of year. It is known that atmospheric ozone over South Africa is mainly impacted by dynamical factors (Bodeker, et al., 2002). Another dynamical factor that influences ozone over the study area is stratospheric-tropospheric exchanges which mostly influence SCO levels. One or a combination of these dynamical factors likely result in low-ozone levels over Cape Point.

**4. Conclusion**

This study evaluated the anti-correlation between ground-based solar UV-B radiation and satellite ozone observations based on clear-sky days at Cape Point, South Africa. The study further investigated whether or not the break-up of the Antarctic ozone hole during spring/summer has an impact on the ozone concentrations over the study area and as a result would have affected solar UV-B radiation levels.

The solar UV-B climatology for Cape Point as well as the climatologies of TOC and SCO followed the expected annual cycle for the Southern Hemisphere. The determination of clear-sky days proved to be reliable in identifying cloudy days. The clear-sky tests removed approximately 87% of days that were affected by cloud cover. At Cape Point, at SZA 20°, an exponential goodness of fit $R^2$ value of 0.710 for TOC with a corresponding RAF value of 0.572 was observed.  At the same SZA, for the correlation of solar UV-B radiation with SCO a goodness of fit $R^2$ value of 0.664 was found.

The break-up of the Antarctic polar vortex had a limited influence on the SCO concentrations Cape Point. The study site was affected to some extent by Antarctic polar air-masses during November months predominately at 600 K and between 475 K and 480 K during certain low-ozone events.  During September months there was less exchange of air-masses between latitudes compared to other months and the study site was affected by midlatitude air-masses during some events. The study site is more

frequently affected by air-masses from tropical regions. The influence of tropical air-masses on the study site is much larger during January and February months. During low SCO events, UVI levels were recorded at ~20% above the climatological monthly mean for events that occurred during September and November.

The relationship between atmospheric ozone and solar UV-B radiation is well understood around the world. The impact of the

Antarctic ozone hole on atmospheric ozone concentrations over South Africa is less well understood. This study showed that there are instances that the Antarctic ozone hole does have a limited effect on ozone concentrations over Cape Point but also shows the effect of tropical air-masses on ozone levels at Cape Point.



**Data availability**

The solar UV-B radiation data are available from the South African Weather Service on request. The total ozone column and stratospheric column ozone data are available on-line from the sources as stated in the manuscript.

**Author contribution**

J.D.P. and C.Y.W. conceived and designed the experiments; J.D.P. performed the experiments; J.D.P. and C.Y.W. analysed the data; J.V.A., H.B. and N.B. contributed to data analysis and interpretation. J.D.P and C.Y.W. wrote the paper.  All authors contributed towards the preparation of the paper.

**Competing interests**

The authors declare that they have no conflicts of interest.

**Acknowledgements**

 The authors would like to thank the South African Weather Service for providing solar UV-B radiation data and cloud cover data. The authors acknowledge the use of Total Ozone Column data from the Ozone Monitoring Instrument (OMI) and Stratospheric Column Ozone data from the Microwave Limb Sounder (MLS).  The following persons are thanked for the various inputs into this project: Dr Greg Bodeker of Bodeker Scientific, funded by the New Zealand Deep South National
Science Challenge, Dr Richard McKenzie, National Institute of Water and Atmospheric Research, New Zealand and Dr Liesl Dyson, University of Pretoria. The University of Reunion is thanked for the provision of the MIMOSA-CHIM model. The SA-French ARSAIO (Atmospheric Research in Southern Africa and Indian Ocean) and PHC-Protea programmes for support for research visits at the University of Reunion. This study was funded in part by the South African Medical Research Council as well as the National Research Foundation of South Africa to grant-holder CY Wright**.**

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
