# Peer review of "Spring and summer time ozone and solar ultraviolet radiation variations over Cape Point, South Africa"

_Annales Geophysicae, 2018_

## Referee Comment (RC1) · Anonymous Referee #1 · 18 Jul 2018

This manuscript describes ozone and solar UV radiation variations over Cape Point. In this manuscript we can see results for South Africa and the author compares some results with Brazil results. Comparison of TOC nad SCO with UV variations shows expectable results. The analysis of the clear sky time series of ozone brings new overview about stratospheric ozone condition on the South hemisphere during the year. My only suggestion is that author should be very careful with definition of clear sky condition because it could strongly affect the results. Next suggestion is that it should be more discussion about results. This manuscript is very descriptive and discussion what phenomena could be responsible for results will improve the paper substantially.

---

## Referee Comment (RC2) · Anonymous Referee #2 · 30 Jul 2018

Considering the topics of this work, one could separate it in two different parts: The first part includes the climatology of UVB and ozone column at Cape Point, South Africa and the corresponding effect of ozone variations on the surface UVB, while the second part includes the examination of low ozone level cases and their origin at the same station. The used methodology and analysis of the second part is an interesting approach with noteworthy findings. The authors could expand their research on this field and publish a standalone study. In my opinion it does not fit in the first part of the manuscript. My major objections to accept for publishing this manuscript concern its first part. Many related studies have been published, especially during 90's, and the results of this study were quite surprising. This fact makes me very doubtful about the

quality and/or the analysis made of the used UVI data. Following, I am quoting some concerns: It is difficult to accept the statement of the authors that there is not aerosol loading at the Cape Point station. Even if the aerosols are not anthropogenic, maritime aerosols affect the site and consequently the UV radiation. As the authors report, the instrument detects solar UV radiation in the wavelength range 280-320 nm. The UV index covers the solar wavelength range 280-400 nm. The conversation of MED to UVI is not just a single factor (equation 1 in the manuscript), but it depends also on the solar zenith angle. There is no information about the long-term performance of the instrument and the calibration procedures during the study period. Taking into account the longitude of the station, the maximum UVI should be observed around 11h00 UTC and not between 13h00-15h00 UTC. Even the presence of the cloudy days is not able to shift the time of the maxima observations 2-4 hours in climatology point of view. If there are any special weather phenomena at the station of Cape Point, (i.e. frequency of clear-skies afternoons much higher than clear-skies mornings) the authors should mention it. The results of RAFs at fixed zenith angles convinced me about my previous concerns. I strongly believe that there are serious mistakes in manipulating the UV data or in the quality of UV data or both. According to my atmospheric physics knowledge, it is impossible the correlation of the RAF at 40 deg to be 0.01 and not significant. I could believe it for SZA higher than 70 deg but not for 40 deg. Even the differences in sun-earth distance could not lead in these results. In case I have some satisfactory answers to my concerns, I will be very happy to accept the manuscript for publication.

---

## Author Response (AR1)

**Response to Reviewers**
**19 September 2018**

**Manuscript title**: Spring and summer time ozone and solar ultraviolet radiation variations over Cape

5  Point, South Africa

Dear Editor,

10  We would like to thank the Reviewers for taking their time to review the manuscript and for providing constructive comments. In the final revised manuscript, we have addressed all of the reviewers' comments. Below we provide a point-by-point response to each of their comments and explain how we revised the manuscript accordingly.

15  **Anonymous Referee #1**

This manuscript describes ozone and solar UV radiation variations over Cape Point. In this manuscript we can see results for South Africa and the author compares some results with Brazil results.

20  Comparison of TOC and SCO with UV variations shows expectable results. The analysis of the clear sky time series of ozone brings new overview about stratospheric ozone condition on the South hemisphere during the year. My only suggestion is that author should be very careful with definition of clear sky condition because it could strongly affect the results. Next suggestion is that it should be more discussion about results. This manuscript is very descriptive and discussion what phenomena

25  could be responsible for results will improve the paper substantially.

**RC1: suggestion is that author should be very careful with definition of clear sky condition because it could strongly affect the results.**
**Response**: We believe that we were sufficiently cautious about our definition of the clear-sky days,

30  and in the revised version of the manuscript we made an effort to state that this is one of the critical

points in the analysis. For that reason, we rephrased parts of Section 2.4.2 on page 6, to draw attention of the reader that: 1) there were no clear-sky observations available for our study site, and 2) hence the need to numerically determine clear-sky days. Our methodology includes three different tests, and we also validated it against measurements for Cape Town, where cloud-cover observations are available.

**RC1: Next suggestion is that it should be more discussion about results**. **This manuscript is very descriptive and discussion what phenomena could be responsible for results will improve the paper substantially.**

**Response**: The full discussion of the phenomena responsible for these results is planned to be presented in a future publication after further research has been carried out. However, in response to the reviewer's comment, the following has been added on page:15, line:14 "In fact, it is well known that Rossby planetary waves are generated due to the development of synoptic disturbances in the troposphere during winter and spring seasons. They propagate vertically through to the stratospheric layers when the zonal winds are westerly (Charney & Drazin, 1961; Leovy, et al., 1985). Moreover, as reported by many authors, gravity and Rossby planetary waves are involved in isentropic transport across the subtropical barrier. Portafaix, et al., 2003 studied the southern subtropical barrier by using MIMOSA model advected PV maps, together with a numerical tool developed by LACy (Reunion University) named DyBaL (Dynamical Barrier Localisation) based on Nakamura formalism (Nakamura, 1996). They showed that the southern subtropical barrier is usually located around 25-30°S, but has an increasing variability during winter and spring. Moreover, using MIMOSA adverted PV fields (Bencherif, et al., 2007; Bencherif, et al., 2003) showed that exchange processes between the stratospheric tropical reservoir and mid-latitudes are episodic and take place through the subtropical barrier due to planetary wave breaking inducing increase or decrease of ozone at tropical and subtropical locations depending on the isentropic levels."

**Anonymous Referee #2**

**General response:** We would like to thank the referee for taking the time to review the manuscript and provide valuable feedback. Clarity and additional information are provided in the responses below.

**RC2: Considering the topics of this work, one could separate it in two different parts: The first part includes the climatology of UVB and ozone column at Cape Point, South Africa and the corresponding effect of ozone variations on the surface UVB, while the second part includes the examination of low ozone level cases and their origin at the same station. The used methodology and analysis of the second part is an interesting approach with noteworthy findings. The authors could expand their research on this field and publish a standalone study. In my opinion it does not fit in the first part of the manuscript.**

**Response:** We thank the reviewer for their comment and appreciate their concern that combining the descriptive part of the study together with analyses used in the 'second part' of the study, however, since very few published studies have appeared since the 1990s that describe the climatology of UV-B and ozone in South Africa, we see value in retaining the first part of our paper here to test contextualise the second half.

**RC2: My major objections to accept for publishing this manuscript concern its first part. Many related studies have been published, especially during 90's, and the results of this study were quite surprising. This fact makes me very doubtful about the quality and/or the analysis made of the used UVI data.**

**Response:** We have completed the re-calculations, we believe that the results are improved. The differences in findings between this study and those from the 1990's is likely due to the different spectral range of instruments used, temporal resolution and specific location.

**RC2: Following, I am quoting some concerns: It is difficult to accept the statement of the authors that there is not aerosol loading at the Cape Point station. Even if the aerosols are not anthropogenic, maritime aerosols affect the site and consequently the UV radiation.**

**Response:** By choosing Cape Point, we assumed that the effect of anthropogenic aerosols on UV radiation was less, say for example in comparison to the City of Cape Town site, but the effect of maritime aerosols is definitely still present. We agree with the referee that in the original version of the paper, we had omitted to be explicit about it, but we made an effort to clearly point this fact in the revised version. For example, in the description of the study area, Section 2.1, page: 3 line: 5, we

added that "…although considered free of air pollution (Slemr, et al., 2008), it may still be affected by maritime aerosols." and later in the same paragraph "… Cape Point offers a setting in which a modification of the UV-B radiation by anthropogenic aerosols can be overlooked." as well as on page: 10, line: 18 "that improvements can be made if the effect of aerosols on UV radiation are considered in future research."

**RC2: As the authors report, the instrument detects solar UV radiation in the wavelength range 280-320 nm. The UV index covers the solar wavelength range 280-400 nm. The conversation of MED to UVI is not just a single factor (equation 1 in the manuscript), but it depends also on the solar zenith angle.**

**Response:** We agree with the reviewer and we have made substantial revisions in this regard. We would like to add J-M. Cadet as co-author for the work done based on these revisions. On page: 4, line: 8 the following has been included" To convert from instrument-weighted UV radiation to erythemally-weighted UV radiation, a correction factor was applied as the instrument does not measure the full spectral range of the UV Index  (Seckmeyer, et al., 2005; Cadet, et al., 2017)."

Figure 2, Equation 1 and 3, Table 2 and 3, the results and discussion have been updated accordingly in the manuscript.

[Figure]

**Figure 1: The *UVI* climatology for all sky conditions at Cape Point. The *x*-axis starts with the month of July and ends with June.**

**Table 2. The correlation statistics for amount of ozone and *UVI* at Cape Point on clear-sky days** (*indicates $R^2$ values were statistically significant at a 95 % confidence interval).

| SZA (°) | TOC: $R^2$ Expo fit | SCO: $R^2$ Expo fit | RAF |
|---|---|---|---|
| 15 | 0.25* | 0.18* | 1.60 |
| 20 | 0.26* | 0.23* | 0.19 |
| 25 | 0.45* | 0.53* | 0.26 |
| 30 | 0.28* | 0.20* | 0.82 |
| 35 | 0.21* | 0.11* | 0.15 |
| 40 | 0.30* | 0.30* | 0.42 |
| 45 | 0.26* | 0.29* | 0.69 |
| **Average** | | | **0.59** |

**Table 3. Identified low-ozone events on clear-sky days at Cape Point during spring and summer months and the percentage decrease calculated from the relative climatological monthly mean** (* indicates whether the low-ozone event was due to low TOC and/or low SCO values).

| Date | TOC (DU) | SCO (DU) | Decrease TOC (%) | Decrease SCO (%) | Increase UVI (%) |
|---|---|---|---|---|---|
| 30 Jan 2009 | 253.5* | 210.4* | 6.1 | 10.1 | 30.4 |
| 6 Feb 2009 | 253.9* | 222.2* | 5.0 | 4.5 | 36.2 |
| 15 Feb 2009 | 254.6* | 228.3 | 4.7 | 1.9 | 34.2 |
| 28 Feb 2011 | 255.7* | 223.2* | 4.3 | 4.1 | 6.8 |
| 16 Jan 2012 | 268.2 | 221.9* | 0.6 | 5.1 | 21.2 |
| 8 Feb 2012 | 257.0* | 227.5 | 3.8 | 2.2 | 31.7 |
| 13 Nov 2012 | 256.6* | 228.8 | 13.3 | 13.3 | 46.5 |
| 14 Nov 2012 | 261.3* | 234.6 | 11.7 | 11.1 | 42.1 |
| 6 Sep 2013 | 265.0* | 241.3* | 12.7 | 11.6 | 22.3 |
| 9 Nov 2013 | 282.3 | 229.0* | 4.6 | 13.3 | 21.9 |
| 1 Sep 2014 | 274.7* | 223.9* | 9.5 | 18.0 | -2.5 |

| | | | | | |
|---|---|---|---|---|---|
| 2 Sep 2014 | 258.4* | 231.2* | 14.9 | 15.3 | -2.3 |
| 9 Sep 2014 | 284.0* | 232.3* | 6.4 | 14.9 | -5.5 |
| 11 Jan 2016 | 270.6 | 221.9* | -0.3 | 5.1 | -7.9 |

**RC2: There is no information about the long-term performance of the instrument and the calibration procedures during the study period.**

**Response**: We acknowledge that the reviewer has made an important comment about the need to express information regarding the calibration of the instruments and the data quality. The following information and additional table have been included in the manuscript regarding the calibration of the Solar Light biometers used at Cape Point on page: 3, line: 16 "Two different instruments were used at Cape Point between 2007 and 2016 (Table 1). The first from January 2007 until March 2016 and the second from April 2016 to December 2016. The SAWS calibrated the instruments at both Solar Light and the Deutscher Wetterdienst (DWD), Lindenberg, Germany. Calibration at Solar Light was according to the "Calibration of the UV radiometer - Procedure and error analysis". At DWD, the instruments were calibrated using the spectrometer SPECTRO 320 D NO 15. During the period of operation for each instrument, the stability was checked by performing inter-comparisons with reference instruments (12010 and 2722) which had been calibrated shortly prior to the inter-comparison.

**Table 1: Summary of the instruments and their calibration information**

| Instrument | Period of Operation | Calibration Information |
|---|---|---|
| Instrument 3719 | January 2007 – March 2016 | Solar Light – June 2006 |
| Instrument 1103 | April 2016 – December 2016 | Solar Light – June 2006 |
| **Inter-comparison Instruments** | **Inter-comparison Date** | **Calibration Information** |
| Instruments 3719 and 12010 | October 2012 | 12010: DWD – August 2012 |
| Instruments 3719 and 2722 | January 2014 | 2722: DWD – July 2013 |

**RC2: Taking into account the longitude of the station, the maximum UVI should be observed around 11h00 UTC and not between 13h00-15h00 UTC. Even the presence of the cloudy days is**

5 **not able to shift the time of the maxima observations 2-4 hours in climatology point of view. If there are any special weather phenomena at the station of Cape Point, (i.e. frequency of clear-skies afternoons much higher than clear-skies mornings) the authors should mention it.**

**Response:** We were further prompted by this referee's comment to look into our tests for determination of clear-sky days and found that Cape Point actually sees more clear-sky afternoons

10 than clear-sky mornings, which could contribute to the shift in the *UVI* maximum. We added this information on page: 8, line: 14 "The maximum *UVI* values are not centred on the local noon, implying that more UV radiation reaches this site in the afternoon. Indeed, as previously mentioned, our clear-sky determination method identified more clear-sky afternoons than clear-sky mornings (Sec. 2.4.2), which under the assumption that cloud cover at Cape Point generally attenuates UV radiation reaching

15 the surface, could explain the observed shift in the *UVI* maximum to about 14h00 SAST." Also, earlier in Section 2.4.2, page: 6, line: 18 "It is interesting to note that at Cape Point, the second test of the clear-sky determination method identified more clear-sky afternoons than clear-sky mornings."

**RC2: The results of RAFs at fixed zenith angles convinced me about my previous concerns. I**

20 **strongly believe that there are serious mistakes in manipulating the UV data or in the quality of UV data or both. According to my atmospheric physics knowledge, it is impossible the correlation of the RAF at 40 deg to be 0.01 and not significant. I could believe it for SZA higher than 70 deg but not for 40 deg. Even the differences in sun-earth distance could not lead in these results.**

25 **Response:** The reviewer has highlighted an important point and we have taken time to consider this and have re-done the RAF calculations. To clarify the results in Table 3, at SZA 40° the RAF value is 0.42. The following has been included on page: 10, line: 8 "At Cape Point, the RAF value for clear-sky days range between 0.15 and 1.60 with an average RAF value of 0.59. This can be interpreted as for every 1% decrease in TOC, UV-B radiation at the surface will increase by 0.59%. RAF values specific

to ozone and solar UV studies found in the literature range between 0.79 and 1.7 (Massen, 2013). RAF values have been used to describe the effect of other meteorological factors such as clouds and aerosols on surface UV radiation (Serrano, et al., 2008; Massen, 2013).  The differences to the RAF values found here and those found in the literature can be attributed to changes in time and location

5   (Massen, 2013). "

**RC2: In case I have some satisfactory answers to my concerns, I will be very happy to accept the manuscript for publication.**

10   **Response:** We made an effort to check our calculations and addressed all the issues raised by the referee. There are a number of instances where we recognised the need to be more specific about our data, as suggested by the reviewer. We hope that the responses given here, and revisions we made to the manuscript will be satisfactory, and that the referee will find the manuscript suitable for publication.

**Spring and summer time ozone and solar ultraviolet radiation variations over Cape Point, South Africa**

D. Jean du Preez[1,2*], Jelena V. Ajtić[3], Hassan Bencherif[4], Nelson Bègue[4], Jean-Maurice Cadet[4] and Caradee Y. Wright[1,5]

[revised manuscript text omitted]

Hourly solar UV-B radiation data were obtained from the South Africa Weather Service (SAWS) for Cape Point station for the period 2007 - 2016. The solar UV-B radiation measurements were made with the Solar Light Model Biometer 501 Radiometer. The biometer measures solar UV radiation with a wavelength of 280 - 320 nm. The measured solar UV radiation is proportional to the analogue voltage output from the biometer with a controlled internal temperature (Solarlight, 2014). Two different instruments were used at Cape Point between 2007 and 2016 (Table 1). The first from January 2007 until March 2016 and the second from April 2016 to December 2016. The SAWS calibrated the instruments at both Solar Light and the Deutscher Wetterdienst (DWD), Lindenberg, Germany. Calibration at Solar Light was according to the "Calibration of the UV radiometer - Procedure and error analysis". At DWD, the instruments were calibrated using the spectrometer SPECTRO 320 D NO 15.

During the period of operation for each instrument, the stability was checked by performing inter-comparisons with reference instruments (12010 and 2722) which had been calibrated shortly prior to the inter-comparison.

**Table 2: Summary of the instruments and their calibration information**

| Instrument | Period of Operation | Calibration Information |
|---|---|---|
| Instrument 3719 | January 2007 – March 2016 | Solar Light – June 2006 |
| Instrument 1103 | April 2016 – December 2016 | Solar Light – June 2006 |
| **Inter-comparison Instruments** | **Inter-comparison Date** | **Calibration Information** |
| Instruments 3719 and 12010 | October 2012 | 12010: DWD – August 2012 |
| Instruments 3719 and 2722 | January 2014 | 2722: DWD – July 2013 |
| Instruments 1103 and 12010 | November 2016 | 12010: Solar Light – June 2015 |

5    Measurements are given in Minimal Erythemal Dose (MED) units where 1 MED is defined by SAWS as 210 $Jm^{-2}$ and any incorrect or missing values were indicated in the dataset. During October 2016, the measured MED values exceeded the expected values and were corrected with a correction factor as recommend by the SAWS. Despite periods of missing data during the study years, there were 3 129 days of useable solar UV-B data for Cape Point. To convert from instrument-weighted UV radiation to erythemally-weighted UV radiation, a correction factor was applied as the instrument does not measure the

10   full spectral range of the UV Index (Seckmeyer, et al., 2005; Cadet, et al., 2017). Solar UV-B radiation values in MED were converted to UV Index (*UVI*) using:

$$UVI = MED[h^{-1}] \frac{210[J \cdot m^{-2}] \times 40 [m^2 \cdot W^{-1}]}{3600[s]} \tag{1}$$

where UVI is the UV Index value and MED is the hourly Minimal Erythemal Dose units.

15   **2.2 Column ozone data**

[revised manuscript text omitted]

$$RAF = \ln\left(\frac{O3}{O3'}\right) / \ln\left(\frac{UVI'}{UVI}\right) \tag{3}$$

where O3 and O3' are the first and second ozone values and $UVI$ and $UVI'$ are the first and second UV measurements, respectively.

**2.4.5 Low-ozone days**

Days of low TOC and SCO values were determined from the set of clear-sky days, but only during spring and summer seasons, when solar UV radiation levels are highest. Days of low TOC values might not have had low SCO values and vice versa. Low TOC and SCO days were determined as days when the respective values were below 1.5 STD from the mean as determined in the climatology analyses (Schuch, et al., 2015).

We then used the MIMOSA-CHIM model to identify whether the origin of ozone-poor air-masses was from the polar region. In other words, on low-ozone days we looked into the maps of advected PV from MIMOSA-CHIM to identify the source of ozone-poor air parcels over the study area.

5 **3. Results and discussion**

**3.1 Climatologies and trends**

**3.1.1 *UVI* climatology**

The monthly means of *UVI* for Cape Point during 2007–2016 were calculated as a function of time of the day and month of the year (Fig. 2). This climatology provides a reliable baseline against which observations can be compared and reveals the
10 general patterns of the *UVI* signal recorded at the surface over the investigated 10-year period.

[Figure]

**Figure 3: The *UVI* climatology for all sky conditions at Cape Point. The *x*-axis starts with the month of July and ends with June.**

At Cape Point, the *UVI* maximum value of approximately 8 UVI occurs between 13h00 and 15h00 South African Standard
15 time (SAST) time, which corresponds to between 11h00 and 13h00 UTC, (Fig. 2). The maximum *UVI* values are not centred on the local noon, implying that more UV radiation reaches this site in the afternoon. Indeed, as previously mentioned, our clear-sky determination method identified more clear-sky afternoons than clear-sky mornings (Sec. 2.4.2), which under the assumption that cloud cover at Cape Point generally attenuates UV radiation reaching the surface, could explain the observed shift in the *UVI* maximum to about 14h00 SAST.

The seasons of maximum (DJF) and minimum (JJA) solar UV-B radiation at Cape Point are as expected for a site in the Southern Hemisphere and are similar to those found in studies at other South African sites, namely Pretoria, Durban, De Aar, and Port Elisabeth and Cape Town (Wright, et al., 2011; Cadet, et al., 2017). The maximum *UVI* values found in this study occur at similar times to Cadet, et al., 2017; Wright, et al., 2011.

**3.1.2 Ozone climatologies**

At Cape Point, TOC (with the maximum of 303.4 DU) and SCO (with the maximum of 273.1 DU) values peaked during September and decreased to a minimum in February for SCO (232.65 DU) and April for TOC (254.49 DU) (Fig. 3). The variations in TOC and SCO are largest at the maximum values and smallest at the minimum values. Over Irene in Pretoria the greatest variation in SCO was seen during spring (Paul, et al., 1998), which is in agreement with our results.  It is suggested that this variability in TOC is due to the movement of midlatitude weather systems which move further north during the Southern Hemisphere winter (Diab, et al., 1992). The climatology of TOC over South Africa is mainly affected by atmospheric dynamics rather than by the effects of atmospheric chemistry (Bodeker & Scourfield, 1998).

[Figure]

**Figure 4: Monthly means and +/- 1.5 STD for total ozone column and stratospheric column ozone starting in July and ending in June.**

The increase in TOC values during the winter months and the maximum during spring months are due to an ozone-rich midlatitude ridge that forms on the equator-side of the polar vortex. The ridge is a result of a distorted meridional flow caused by the Antarctic polar vortex that forms in late autumn. The vortex prevents poleward transport of the air, and thus allows for a build-up of ozone-rich air in midlatitudes. The lower TOC values over summer could be due to the dilution effect of ozone-poor air from the Antarctic ozone hole. The dilution effect occurs when the vortex breaks-up (Bodeker & Scourfield, 1998; Ajtić, et al., 2004).

**3.2 Correlation between the amount of ozone and *UVI**

The first order exponential goodness of fit $R^2$ values at fixed SZAs (Table 2) describe the anti-correlation between the amount of ozone in the atmosphere and *UVI*. The strongest anti-correlation was found at a fixed SZA 25° for both TOC and SCO. In this study the first order exponential fit was used to describe the anti-correlation between ozone and solar UV radiation as in some instance this is best described with a non-linear fit (Guarnieri, et al., 2004). A study on the anti-correlation between solar UV-B irradiance and TOC in southern Brazil found that the percentage of the $R^2$ values for exponential fits (66.0 -85.0%)

explained the variations in solar UV-B irradiance due to TOC variations on clear-sky days at the same fixed SZA categories used in our study (Guarnieri, et al., 2004).

**Table 2. The correlation statistics for amount of ozone and *UVI* at Cape Point on clear-sky days** (*indicates $R^2$ values were statistically significant at a 95 % confidence interval**).**

| SZA (°) | TOC: $R^2$ Expo fit | SCO: $R^2$ Expo fit | RAF |
|---|---|---|---|
| 15 | 0.25* | 0.18* | 1.60 |
| 20 | 0.26* | 0.23* | 0.19 |
| 25 | 0.45* | 0.53* | 0.26 |
| 30 | 0.28* | 0.20* | 0.82 |
| 35 | 0.21* | 0.11* | 0.15 |
| 40 | 0.30* | 0.30* | 0.42 |
| 45 | 0.26* | 0.29* | 0.69 |
| **Average** | | | **0.59** |

The exponential $R^2$ values of TOC found at Cape Point at a fixed SZAs were much lower than those southern Brazil. In this study and in other studies the $R^2$ value is smaller sat the largest SZAs (Guarnieri, et al., 2004; Wolfram, et al., 2012).

At Cape Point, the RAF value for clear-sky days range between 0.15 and 1.60 with an average RAF value of 0.59. This can be interpreted as for every 1% decrease in TOC, UV-B radiation at the surface will increase by 0.59%. RAF values specific to ozone and solar UV studies found in the literature range between 0.79 and 1.7 (Massen, 2013). RAF values have been used to describe the effect of other meteorological factors such as clouds and aerosols on surface UV radiation (Serrano, et al., 2008; Massen, 2013). The differences to the RAF values found here and those found in the literature can be attributed to changes in time and location (Massen, 2013).

[revised manuscript text omitted]

**3.4 Origin of ozone-poor air**

In this section the model results from MIMOSA-CHIM are shown for a selection on low-ozone events. The latitude origin of air masses was classified according to the colour scale on the PV maps. Blue colours indicate air-masses with relatively high PV values, implying their polar origins, while red colours indicate relatively low PV values of tropical origin.

The origin of the air-masses for low-ozone events in January (Table 3) and February (Table 4) shows a consistent pattern: in the lower parts of the stratosphere, at 425 K, the air was of tropical origin; higher up, at 475 K, of midlatitude origin; and at 600 K, air masses from the polar region were above Cape Point. This pattern in illustrated in the PV maps from MIMOSA-CHIM of the low-ozone event on 16 January 2012 (Fig. 5), which best demonstrates all January events. During this event, we identified low SCO (Table 2). Similarly, the PV maps from MIMOSA-CHIM for the low-ozone event on 6 February 2009 (Fig. 6) best demonstrate the situation for February months.

Our results imply that the low-ozone events during the months of January and February were not directly influenced by the Antarctic ozone hole as by that time, the polar vortex is already broken up. However, it is possible that these events are a consequence of the ensuing mixing of the polar ozone-poor air that reduces the midlatitudes ozone concentrations (Ajtić et al. 2004).

[revised manuscript text omitted]

Our results imply that the break-up of the Antarctic polar vortex has a limited influence on the SCO concentrations over Cape Point. The study site was affected to some extent by Antarctic polar air-masses during November months predominately at 600 K. During September low-ozone events, there was less exchange of air-masses between latitudes compared to other months and the study site was mostly under the influence of midlatitude air-masses. The study site seems to be more frequently affected by air-masses from the tropical regions, especially in the lower stratosphere. Further, the influence of tropical air-masses on the study site is larger during January and February months. During low SCO events in September and November, the recorded *UVI* levels were ~20 % above the climatological monthly mean.

The relationship between atmospheric ozone and solar UV-B radiation is well understood around the world. The impact of the Antarctic ozone hole on atmospheric ozone concentrations over South Africa is less well understood. Our study showed instances when the Antarctic ozone hole seems to have a limited effect on ozone concentrations over Cape Point but also showed the effect of tropical air-masses on ozone levels at Cape Point.

**Data availability**

The solar UV-B radiation data are available from the South African Weather Service on request. The total ozone column and stratospheric column ozone data are available on-line from the sources as stated in the manuscript.

**Author contribution**

J.D.P. and C.Y.W. conceived and designed the experiments; J.D.P. performed the experiments; J.D.P. and C.Y.W. analysed the data; J.M.C assisted with data conversion. J.V.A., H.B. and N.B. contributed to data analysis and interpretation. J.D.P and C.Y.W. wrote the paper. All authors contributed towards the preparation of the paper.

**Competing interests**

The authors declare that they have no conflicts of interest.

**Acknowledgements**

The authors would like to thank the South African Weather Service for providing solar UV-B radiation data and cloud cover data. The authors acknowledge the use of Total Ozone Column data from the Ozone Monitoring Instrument (OMI) and Stratospheric Column Ozone data from the Microwave Limb Sounder (MLS). The following persons are thanked for the various inputs into this project: Dr Greg Bodeker of Bodeker Scientific, funded by the New Zealand Deep South National Science Challenge, Dr Richard McKenzie, National Institute of Water and Atmospheric Research, New Zealand and Dr Liesl Dyson, University of Pretoria. The University of Reunion is thanked for the provision of the MIMOSA-CHIM model and the CCUR team for the use of the TITAN supercomputer. The SA-French ARSAIO (Atmospheric Research in Southern Africa and Indian Ocean) and PHC-Protea programmes for support for research visits at the University of Reunion. This study was funded in part by the South African Medical Research Council as well as the National Research Foundation of South Africa to grant-holder CY Wright.